# Anticancer, Immunomodulatory, and Phytochemical Screening of *Carthamus oxyacantha* M.Bieb Growing in the North of Iraq

**DOI:** 10.3390/plants13010042

**Published:** 2023-12-22

**Authors:** Media Mohammad Baban, Saman A. Ahmad, Ala’ M. Abu-Odeh, Mustafa Baban, Wamidh H. Talib

**Affiliations:** 1Department of Clinical Pharmacy and Therapeutics, School of Pharmacy, Faculty of Pharmacy, Applied Science Private University, Amman 11931-166, Jordan; mediababan22@gmail.com; 2Biotechnology and Crop Science Department, College of Agriculture Engineering Science, University of Sulaimani, Sulaimani 46001, Iraq; saman.ahmad@univsul.edu.iq; 3Botanical Foundation, The American University of Iraq, Sulaimani 46001, Iraq; 4Department of Pharmaceutical Chemistry and Pharmacognosy, School of Pharmacy, Applied Science Private University, Amman 11931-166, Jordan; a_abuodeh@asu.edu.jo; 5Department of Medicine and Surgery, School of Medicine, Campus of St George’s University of London, Cranmer Terrace, London SW17 0RE, UK; mustafababan@doctors.org.uk; 6Faculty of Allied Medical Sciences, Applied Science Private University, Amman 11931-166, Jordan

**Keywords:** *Carthamus oxyacantha* M.Bieb, LC-MS, phytochemicals, anticancer, immunomodulatory, antioxidant activity

## Abstract

*Carthamus oxyacantha* M.Bieb is a promising repository of active phytochemicals. These bioactive compounds work synergistically to promote the plant’s antioxidant, anticancer, and immunomodulatory capabilities. The present study aimed to discover the potential immunomodulatory and cytotoxicity of different extracts of *Carthamus oxycantha* roots. Aqueous ethanol (70%), aqueous methanol (90%), ethyl acetate, and n-hexane extracts were tested against five cell lines (T47D, MDA-MB231, Caco-2, EMT6/P, and Vero). Among these extracts, ethyl acetate and n-hexane extracts showed significant activity in inhibiting the proliferation of cancerous cells because of the presence of several phytochemical compounds, including flavonoids, phenolics, and alkaloids. The n-hexane extract was the most potent extract against T47D and Caco-2 cell lines and had IC_50_ values of 0.067 mg/mL and 0.067 mg/mL, respectively. In comparison, ethyl acetate extract was active against T47D and MDAMB231, and IC_50_ values were 0.0179 mg/mL and 0.03 mg/mL, respectively. Both n-hexane and ethyl acetate extracts reduced tumor size (by 49.981% and 51.028%, respectively). Remarkably, *Carthamus oxyacantha* extracts decreased the average weight of the tumor cells in the in vivo model. The plant induced significant apoptotic activity by the activation of caspase-3, immunomodulation of macrophages, and triggering of pinocytosis. The implications of these intriguing findings demand additional research to broaden the scope of the understanding of this field, opening the doors to the possibilities of using *Carthamus oxyacantha* M.Bieb as an effective cancer treatment adjuvant in the future.

## 1. Introduction

Cancer remains a formidable global health challenge that continues to impact individuals worldwide. Cancer significantly affects developed and developing countries [1]. In 2019, cancer was positioned as the first and second primary cause of death in people of age ≤ 70 years in 112 out of 183 countries [2]. In 2020, around 10 million cancer-related deaths were reported, and cancer was positioned as the third and fourth leading cause of mortality in 23 countries [3]. These alarming statistics from the World Health Organization (WHO) and other reputable studies highlight the critical need to investigate novel therapeutic avenues to combat this intractable disease.

The pathophysiology of cancer is extraordinarily complex, involving a wide variety of molecular pathways and risk factors [4]. Cancer cells acquire the ability to proliferate uncontrollably, infiltrate into the adjacent tissues, and disseminate via metastasis upon acquiring mutations [5]. Breast cancer stands out as the most frequently diagnosed cancer in females, and up to 20–30% of the females can develop a metastatic disease that frequently spreads into bones and other distant organs, while colorectal cancer remains the third most commonly diagnosed cancer and the second primary cause of cancer-related mortality worldwide [6,7].

Current cancer therapies encompass a vast array of therapeutic modalities that are tailored to the type, stage, and severity of cancer in each patient [8]. Concerns regarding drug resistance and adverse effects associated with conventional therapies, namely chemotherapy, have compelled researchers to investigate alternative approaches [9]. In this context, natural products have emerged as promising sources of potential anticancer agents with reduced toxicity.

Several medicinal plants have been explored which have been used for the treatment of various ailments in traditional medicine for centuries and offer a balanced immune function, which is essential for protecting the body against antigenic invasion, mainly by microorganisms [10,11]. In complex interactions with immune cells and cytokines, these natural immunomodulatory substances from medicinal plants have been shown to enhance the body’s immune response to diseases [12]. Immunomodulators play pivotal roles in supporting immune function, either stimulating or suppressing the immune system’s response to attacking pathogens. These natural immunomodulators can be used as safer alternatives to clinically used immunosuppressant and immunostimulant cytotoxic drugs which possess severe side effects. Many plants of different species have been reported to possess strong immunomodulating properties. In this pursuit of finding new therapeutics, paramount importance rests in the exploration of natural resources [11].

*Carthamus oxyacantha*, commonly known as “wild safflower”, is one of the largely unexplored gems in the domain of natural products [13]. This plant has traditionally been used for wound healing and anti-inflammatory purposes, but its immunomodulatory and anticancer properties remain to be elucidated. With its abundance of phenolic compounds, flavonoids, sterols, and other active constituents, *Carthamus oxyacantha* offers a promising opportunity for the development of novel pharmacological medicines for cancer treatment [14].

This investigation seeks to shed light on the immunomodulatory and anticancer properties of root extracts of *Carthamus oxyacantha* M.Bieb, as well as to elucidate its various phytochemical constituents. By doing so, we aim to uncover the therapeutic capability of this plant and its potential applications in cancer medicine by conducting an extensive investigation into its various characteristics and attributes.

## 2. Materials and Methods

### 2.1. Instruments, Reagents, and Commercial Kits

The equipment, reagents, and kits that were utilized in the current study were CO_2_ incubator model No. 460-1CE (Lab-Line Instruments Incorporation, Kerper Blvd, Dubuque, IA, USA), water bath model No. 1083 (GFL, Berlin, Germany), a hemocytometer (Neubauer, Erzhausen, Germany), a lyophilizer (Edwards, Burgess Hill, UK), a rotary evaporator (Heidolph WB 2000, Heidolph Instruments GmbH & Co. KG, Schwabach, Germany), UV spectrophotometer Cary 8454 UV–visible spectrophotometer (Agilent Technologies, Woburn, MA, USA), Ultrasonication bath—Analogic ultrasonic bath Mod. AU-220 (AgroLab, Carpi, Italy). RPMI 1640 (Roswell Park Memorial Institute medium), MEM (minimum essential medium), DMEM (Dulbecco’s modified Eagle medium), FBS (fetal bovine serum), L-glutamine, penicillin–streptomycin solution, gentamycin, trypan blue 0.4%, PBS (phosphate-buffered saline), and trypsin–EDTA (ethylenediamine tetraacetic acid) were purchased from Sigma, St. Louis, MO, USA. Concanavalin A (Con A) (Santa Cruz Biotechnology, Dallas, TX, USA) and lipopolysaccharides from Escherichia coli (LPS) (Sigma, USA) were utilized to evaluate lymphocyte proliferation. Gallic acid, DPPH (2,2-diphenyl-1-picrylhydrazyl), and Trolox were purchased from Sigma-Aldrich (Stenheim, Germany). An MTT (3-(4, 5-Dimethylthiazol-2-yl)-2, 5-diphenyltetrazolium bromide) kit (Bioworld, London, UK) was used for the antiproliferative assay. An enzyme-linked immunosorbent assay (ELISA) kit (Invitrogen, Vienna, Austria) was used for the quantitative detection of human caspase-3.

### 2.2. Carthamus oxyacantha Supply and Extracts Preparation by Maceration

The plant identification was conducted by assistant Prof. Saman A. Ahmed. The plant was collected from May to July 2022 from the hills and mountains of Sulaimaneyah, Iraq. The plant samples were cleaned thoroughly with water. Roots were separated, cut into small pieces, and left to dry outdoors in the shade for approximately 5 days in a row with continuous flipping over. The dried roots were ground using an electric mixer grinding machine. The plant extracts were obtained using solvents of various polarities. *Carthamus oxyacantha* (100 g in 1 L) was macerated in aqueous ethanol (70%), aqueous methanol (90%), ethyl acetate, and n-hexane for 14 days at room temperature (25 °C) with continuous stirring. Following the removal of the residue, the supernatant was vacuum filtered, then concentrated using a rotary evaporator and lyophilized. The extracts were stored at −20 °C until further use.

### 2.3. Carthamus oxyacantha Extracts Preparation Using the Soxhlet

A Universal Extractor device (Buchi, mod. E-800, Uster, Switzerland) was used to isolate the extracts from 200 g of the plant using the solvents that had the lowest yield, which were ethyl acetate and n-hexane. These solvents were used for extraction in the ratio of 10:1. After 4 h of continuous extraction, samples were washed for 5 min, followed by 5 min of solvent evaporation. Using a rotatory evaporator, the extracts were dried out while keeping the bath temperature at 40 °C.

### 2.4. Cell Lines and Cell Culturing Condition

Five cell lines (T47D, MDA-MB231, Caco-2, EMT6/P, and Vero) were chosen to explore the possible anticancer impact of *Carthamus oxyacantha* extracts. They were obtained from the American Type Culture Collection (ATCC, Manassas, VA, USA). Doxorubicin was utilized as a positive control, obtained from EBEWE Pharma (Unterach am Attersee, Austria). In addition, the complete tissue culture medium was prepared by supplementing it with 10% fetal bovine serum, 1% penicillin–streptomycin solution, 1% L-glutamine, 0.1% non-essential amino acids, and 0.1% gentamycin solution. To 500 mL bottles of media (as supplied), ingredients were added as 50 mL, 5 mL, 5 mL, 0.5 mL, and 0.5 mL, respectively. The T47D cell line was cultivated in RPMI 1640, whereas the EMT6/P cell line was cultivated in MEM, and the remaining cell lines were cultivated in complete DMEM high-glucose media. Cells were incubated at 37 °C in a CO_2_ incubator up to 80–90% confluence [15].

### 2.5. Antiproliferative Assay

The antiproliferation activity of the extracts was assessed using MTT assay (3-(4,5-Dimethylthiazol-2-yl)-2,5-diphenyltetrazolium bromide) test. Cells were adjusted to 1.5 × 10^4^/mL in each well of the 96-well cell culture plate (Biofil, Guangzhou, China). After incubation at 37 °C in an incubator containing 5% CO_2_ and 95% humidity, the cells were treated with various concentrations of *C*. *oxyacantha* extracts (0.078–5 mg/mL). After 1 day or 2 days of incubation, cell viability was assessed using the MTT test kit (Bioworld, UK) using a procedure adopted from Alobaedi [15]. For this purpose, 10 μL of thiazolyl blue tetrazolium solution was added to each well, and the plate was incubated for 3 h. Optical density (OD) was measured using the ELISA microplate absorbance reader (BioTek, Highland Park, IL, USA) at 550 nm. The IC_50_ values and the percentage of surviving cells were determined using SPSS 25.
$$Percentage\;of\;Cell\;Viability\left(\%\right)=\left(OD\;of\;treated\;\frac{cell}{OD}\;of\;control\;cell\right)*100$$

Doxorubicin (Dox) possesses a broad spectrum of anticancer activity [16]. Doxorubicin was used as the positive control in this study.

The estimation of the selectivity index (SI) was based on the ratio of IC_50_ of Vero cells to IC_50_ of the other cancerous cell lines. The samples were classified as highly selective if their SI value exceeded 3 [17].
SI=IC50 of Vero cell line/IC50 of each extract on different cancer cell line

### 2.6. Apoptosis Detection Assay

Cells were cultured in four tissue culture flasks at a concentration of 1.5 × 10^4^ cells/mL and incubated for 24 h. Following incubation, the cells were subjected to one of three treatments of *Carthamus oxyacantha* extracts: aqueous ethanol (70%) extract at a concentration of 4 mg/mL; n-hexane extract at a concentration of 0.5 mg/mL; ethyl acetate extract at a concentration of 0.014 mg/mL. For the positive control sample, 250 nM of doxorubicin hydrochloride was added, and for the negative control sample, only the medium was added. The cells were incubated for 48 h at 37 °C in a CO_2_ incubator up to 80–90% confluence. Subsequently, cells from the media were collected. The measurement of caspase-3 activity was conducted following the manufacturer’s protocol and instructions provided in the kit (Caspase 3 Assay Kit, Colorimetric) obtained from Sigma-Aldrich (USA) and the fold increase in caspase-3 activity was assessed.

### 2.7. Liquid Chromatography–Mass Spectrometry (LC-MS) Assay Results Indicate That the Ethyl Acetate Extract, at a Concentration of 10 mg/mL, Demonstrated the Highest Level of Peritoneal Phagocytic Activity

Freshly prepared ethyl acetate and n-hexane samples were prepared by dissolving the extract in the solvent. Samples were filtered through a 0.45 μm membrane filter (cellulose acetate (CA) membranes made of entirely cellulose acetate polymer, Sterlitech, Auburn, WA, USA). All samples were prepared and used immediately for analysis. Extracts were analyzed using the Exion LC system (SCIEX, Framingham, MA, USA), fitted with an X500 QTOF mass spectrometer (SCIEX) equipped with an ESI. Separation was performed on an InertSustain C18 (GL Sciences Inc., Tokyo, Japan, 25 cm × 4.6 mm × 5 m). The mobile phase was with (A) 0.1% formic acid in water (1:1000) and (B) acetonitrile with a gradient flow rate of 1.0 mL/min. The injection volume was set to 0.6 μL. Positive-mode LC-MS was performed for which the ion spray voltage was 5000 V, the declustering potential was 80 V, and the collision energy was 10 V.

### 2.8. Animals

The Research and Ethical Committee of the Faculty of Pharmacy at the Applied Science Private University approved all of the experimental protocols used in this study, which was conducted in accordance with generally accepted ethical standards, with a reference number of 2022-PHA-37. Balb/C female mice between the ages of 4–6 weeks and weighing 23–25 g were used for this study to perform in vivo analysis. The animals were maintained under standard conditions, temperature of 24–25 °C, 50–60% relative humidity, with water, food, and a sterile diet in a pathogen-free environment and maintained on a 12 h light/dark cycle.

### 2.9. Preparation of Murine Splenocytes

The female Balb/C mice were sacrificed and their spleens were harvested. The aseptic extraction of the splenocytes was performed using sterilized tools, and the cells were kept in Petri dishes, with 5 mL of RPMI, 0.5 penicillin, and 0.5 gentamycin. The cell suspension was centrifuged for 5 min at a speed of 225 g using a refrigerated centrifuge (MPW-260R, Warsaw, Poland). Cells were then treated with RBC lysis buffer (NH₄CL RBC lysis buffer for human (Bio-World, Visalia, CA, USA)) to remove the red blood cells, and splenocytes were counted and seeded for the next experiments.

### 2.10. Lymphocyte Proliferation Assay

Lymphocyte proliferation assay was performed in the presence of the mitogens Con A (concanavalin A) (5 μg/mL) and LPS (lipopolysaccharide) (4 μg/mL). These were used as mitogens for T and B lymphocytes, respectively. Splenocyte suspension was prepared (2 × 10^6^ cells/mL) in RPMI1640 supplemented with 50 U/mL penicillin, 50 U/mL streptomycin, and 10% FBS, and then it was seeded into a 96-well culture plate containing either 5μg/mL Con A or 4 μg/mL LPS. Then, 100 μL of (5–20 mg/mL) of the extracts was added in triplicates. Moreover, the negative control contained the same volume of RPMI 1640 medium. The plate was incubated for 48 h under 5% CO_2_ and a humidified atmosphere at 37 °C temperature. Afterward, 10 μL of (5 mg/mL) MTT solution [3-(4, 5-Dimethylthiazol-2-yl)- 2, 5- diphenyltetrazolium bromide] was added to each well and incubated for about 4 h, followed by 100 μL DMSO. The absorbance was measured with an ELISA microplate reader at 570 nm. Results were expressed as a percentage of proliferation (%) compared with the negative control cells. The same technique was repeated without Con A and LPS [18].
% Proliferation=OD test−OD control∗100/OD control

### 2.11. Macrophage Isolation from Peritoneal Fluid

Mice were injected intraperitoneally with 5 mL of 3% (*w*/*v*) Brewer’s thioglycollate medium 48 h before the collection of peritoneal macrophages (PEM). Peritoneal macrophages were isolated from the peritoneal cavity as described previously [19]. The cells were centrifuged at 2000 RPM and resuspended in Roswell Park Memorial Institute (RPMI) in 1640 medium (Euroclone SpA, Milan, Italy).

### 2.12. Phagocytic Activity Assay

The nitro blue tetrazolium (NBT) reduction test was performed following the procedures outlined by [20]. In a 96-well plate, peritoneal macrophages were seeded at a concentration of 5 × 10^6^ cells/well and then cultivated at 37 °C for 48 h with varying doses of *Carthamus oxyacantha* extracts (10–1.5 mg/mL). Next, 20 µL of yeast suspension (5 × 10^7^ cells/mL in PBS) and 20 µL of nitro blue tetrazolium (NBT) (1.5 mg/mL in PBS) were added to each well, except for the control wells, which received just yeast suspension (5 × 10^7^ cells/mL in PBS). The supernatant was discarded after 60 min of incubation, and the macrophages were treated with 140 µL of DMSO and 120 µL of 2M KOH. Finally, the OD was obtained at 550 nm in the microplate reader. Phagocytic activity was calculated based on the following equation:$$Phagocytic\;index=\left(OD\;sample-OD\;control\right)*100/OD\;control$$

### 2.13. Pinocytic Activity Assay

To evaluate the potential of *Carthamus oxyacantha* extracts on macrophages’ pinocytic activity, the neutral red technique was adopted from Boothapandi et al. and Phytycz et al. [20,21]. Peritoneal macrophages were seeded into a 96-well plate (5 × 10^6^ cells/well). After being treated with various concentrations of the plant extracts (10–1.5 mg/mL), the plate was incubated at 37 °C for 48 h. Afterward, 100 μL of a neutral red solution was added to each well, and they were incubated for 2 h, after which cells were washed with PBS. Then, 100 μL of cell lysis solution (ethanol and 0.01% acetic acid in a 1:1 ratio) was added to each well to lyse the cells. The plate was incubated overnight at room temperature, and OD was determined at 550 nm. Absolute OD values representing dye uptake were used to measure pinocytic activity.

### 2.14. Acute Toxicity Test of Carthamus oxyacantha M.Bieb Extract to Determine the LD50

A study was conducted on a small group of mice to determine the median lethal dose (LD50). N-hexane and ethyl acetate extracts of *Carthamus oxycanthus* were dissolved in PBS and 5% Tween 20. Five female mice (6 weeks, 20–23 g) were injected intraperitoneally (IP) with 0.8 g/kg of plant extract. They were monitored for 24 h for any signs of mortality. The next day’s dose was adjusted by increasing it by 1.5-fold if it was well tolerated or decreasing it by 75-fold if it was toxic. The maximum non-lethal and minimum lethal concentrations were used as the minimum and maximum limits for calculating LD50 doses. Five groups of mice (*n* = 6) were injected intraperitoneally with doses of 1200, 1500, 1600, 1700, and 1800 mg/kg. The sixth group was assigned as control and was administered PBS. The mice were observed for 24 h to assess mortality rate. The LD50 was determined using the Karber method [22].

### 2.15. Antitumor Effect of Carthamus oxyacantha M.Bieb on Mice Model

A cohort of 24 Balb/C mice was divided into 3 groups, with 6 mice in each group. Group I: positive control group (healthy mice fed usual fodder and inoculated with tumors); Group II: n-hexane extract group; and Group III: ethyl acetate extract group. For the tumor inoculation, 100 μL (1.5 × 10^4^ cells) of EMT-6/P cells were injected subcutaneously into the abdomen region of each mouse, followed by a wait of 10 days for the tumor to develop. After 7 days, mice received daily IP injections of *Carthamus oxyacantha* n-hexane and ethyl acetate extracts at 182 mg/kg. Till 19 days, the size of the tumors was determined by measuring their length and width with a digital caliper [23]. The following formula was utilized in the estimation of tumor volumes:V=L∗W∗W∗0.5
where V, L, and W are the volume, length, and width of the tumor, respectively.

### 2.16. Evaluation of Liver and Kidney Function in Treated Mice

Serum samples were collected from the mice and tested for aspartate transaminase (AST), alanine transaminase (ALT), and creatinine using commercially available kits following the manufacturer’s instructions (BioSystems, Barcelona, Spain). Serum levels of the liver enzymes AST and ALT were measured in mice treated with *Carthamus oxyacantha*, untreated, and tumor-free normal mice, and absorbance was determined using a spectrophotometer set at 340 nm. Serum creatinine levels were also analyzed for the same groups, and absorbance was measured at 500 nm. Working reagents (supplied with the kit and used in sample preparation) were utilized as blanks.

### 2.17. Statistical Analysis

Data were represented using the mean ± standard deviation of triplicate independent experiments utilizing the GraphPad prism^®^ version 9.5.1 for the statistical analysis by performing one-way analysis of variance (ANOVA) and post hoc analysis using Dunnett’s multiple comparisons test to investigate the significant differences in the mean values for each parameter between the different groups of the experiment. The differences between groups’ mean values were considered significant when *p* < 0.05. IC_50_ values (the concentrations at which there were 50% cell death compared with the negative control) were assessed using nonlinear regression in SPSS (Statistical Package for the Social Science, Chicago, IL, USA, version 22).

## 3. Results

### 3.1. Antiproliferative Activity of Carthamus oxyacantha M.Bieb Different Extracts on Different Cell Lines

In a dose-dependent manner, decreasing the concentration of *Carthamus oxyacantha* extracts on the MDA-MB231 cell line resulted in an increase in the average percentage of cell survival. The concentration ranged between 5 to 0.078 mg/mL. At a concentration of 5 mg/mL, the percentages of inhibition caused by aqueous ethanol (70%), aqueous methanol (90%), n-hexane, and ethyl acetate extracts were 57.8%, 52.8%, 60.7%, and 65.3%, respectively. N-hexane and ethyl acetate extracts showed the highest activity, with IC_50_ values of 0.263 mg/mL and 0.03 mg/mL, respectively. In contrast, aqueous ethanol (70%) and aqueous methanol (90%) extract had the least effectiveness against this cell line, with IC_50_ values of 0.534 mg/mL and 2.98 mg/mL, respectively (Figure 1, Figure 2, Figure 3 and Figure 4). For the T47-D cell line at the concentration of 5 mg/mL, the percentages of inhibition caused by aqueous ethanol (70%), aqueous methanol (90%), n-hexane, and ethyl acetate extracts were 76%, 42%, 71.7%, and 68%, respectively. N-hexane and ethyl acetate extracts showed the highest activity, with IC_50_ values of 0.067 mg/mL and 0.0179 mg/mL, respectively. In contrast, aqueous ethanol (70%) and aqueous methanol (90%) extracts had the least effectiveness against this cell line, with IC_50_ values of 0.691 mg/mL and more than 5 mg/mL for the aqueous methanol extract (90%) (Figure 1, Figure 2, Figure 3 and Figure 4). At a concentration of 5 mg/mL, the percentages of inhibition of EMT6/p cells caused by aqueous ethanol (70%), aqueous methanol (90%), n-hexane, and ethyl acetate extracts were 49%, 25%, 56%, and 58%, respectively. N-hexane and ethyl acetate extracts showed the highest activity, with IC_50_ values of 0.398 mg/mL and 0.497 mg/mL, respectively, whereas aqueous ethanol (70%), and aqueous methanol (90%) extracts had the least effectiveness against this cell line, with IC_50_ values of 0.816 mg/mL and more than 5 mg/mL for the aqueous methanol extract (90%) (Figure 1, Figure 2, Figure 3 and Figure 4). Similarly, for the Caco-2 cell line, at the extract concentration of 5 mg/mL, the percentages of inhibition caused by aqueous ethanol (70%), aqueous methanol (90%), n-hexane, and ethyl acetate extracts were 69%, 54%, 76%, and 52%, respectively. N-hexane and ethyl acetate extracts showed the highest activity, with IC_50_ values of 0.067 mg/mL and 0.214 mg/mL, respectively, whereas aqueous ethanol (70%) and aqueous methanol (90%) extracts had the least effectiveness against this cell line, with IC_50_ values of 0.497 mg/mL and more than 5 mg/mL for the aqueous methanol extract (90%) (Figure 1, Figure 2, Figure 3 and Figure 4). The antiproliferative trials used doxorubicin as a positive control. The IC_50_ values of doxorubicin against the T47d, EMT6/P, MDA-MB231, Caco-2, and Vero cell lines were 7.01, 0.57, 15.56, 0.47, and more than 200 mg/mL, respectively. Using five cell lines, Table 1 shows the IC_50_ for all extracts and doxorubicin.

The selectivity index (SI) was also determined for aqueous ethanol (70%), aqueous methanol (90%), n-hexane, and ethyl acetate extracts of *Carthamus oxyacantha* against T47d, EMT6/P, MDA-MB231, and Caco-2 cell lines to determine the safe active concentration for each extract and compare it with the Vero normal cell line concentration. Table 2 lists the SI of the plant extracts that were assessed.

### 3.2. Comparison between the IC_50_ Values of Various Carthamus oxyacantha Extracts Tested on T47d, EMT6/P, MDA, Caco-2, and Vero Cell Lines

The half maximal inhibitory concentration (IC_50_) is the concentration of a substance required to induce 50% cell death relative to negative control. Table 3 shows the IC_50_ values for each extract when treated with each cell line, beginning with the n-hexane extract, which was the most active extract against T47D and Caco-2 cell lines with corresponding IC_50_ values of 0.067 mg/mL and 0.067 mg/mL respectively. Ethyl acetate extract also exhibited significant inhibitory activity against T47D and MDA-MB231 cells, with IC_50_ values of 0.0179 mg/mL and 0.03 mg/mL, respectively. It was also revealed that the aqueous ethanol extract (70%) had a slight inhibitory activity against all cell lines. However, the last extract, aqueous methanol (90%), did not show any significant effect against any cell line.

For the Vero cell line, these extracts (aqueous ethanol (70%), aqueous methanol (90%), n-hexane, and ethyl acetate) showed some noticeable toxicity, with IC_50_ values of 0.753 mg/mL, 4.7 mg/mL, 0.46 mg/mL, and 0.251 mg/mL, respectively.

### 3.3. LC-MS Analysis

LC-MS analysis of ethyl acetate and hexane extract identified the presence of various compounds of different phytochemical classes, such as phenolics, flavonoids, alkaloids, terpenes, and coumarins. Table 3 shows phytochemical compounds found in the ethyl acetate and n-hexane extracts of *Carthamus oxyacantha* M.Bieb using LC-MS.

### 3.4. Lymphocyte Proliferation Assay

The ethyl acetate extract exhibited the highest response in the lymphocyte proliferation assay at a concentration of 10 mg/mL compared with the control, with *p* < 0.0001, both in the presence and absence of mitogens. The three other extracts also exhibited comparable statistical significance at an equivalent concentration, with *p* ≤ 0.0001. The ethyl acetate extract (10 mg/mL) showed a stimulation index (proliferation index) of approximately 12.15 and 11.05 in cells stimulated with Con A and LPS, respectively. However, the percentage proliferation of ethyl acetate extracts (10 mg/mL) under mitogen-free conditions was 10.20; it is noteworthy that the presence of Con A resulted in the most significant impact across all the extracts, as depicted (Figure 5, Figure 6 and Figure 7).

### 3.5. The Effect of Different Carthamus oxyacantha Extracts on Phagocytosis and Pinocytosis

The assay results indicate that the ethyl acetate extract at a concentration of 10 mg/mL demonstrated the highest level of peritoneal phagocytic activity, with a phagocytic index of (390%) (Figure 8a). In comparison, the control group exhibited a phagocytic index of 0.108%. The aqueous methanol extract (90%) also induced comparable peritoneal phagocytic activity at concentrations of 10, 5, and 2.5 mg/mL. At a concentration of 10 mg/mL, aqueous methanol extract (90%) exhibited a significant level of phagocytic activity with a phagocytic index of 238%. Nevertheless, all the extracts demonstrated statistical significance (*p* < 0.0001) across the four selected concentrations (Figure 8a).

In order to provide an estimation of the effect of each solvent extract, the pinocytotic activity of macrophages was assessed. The pinocytotic activity of the cells was enhanced when they were treated with extracts at dosages ranging from 10 to 1.25 mg/mL. The most significant increase in the pinocytotic activity was induced by the aqueous ethanol extract (70%), which had a pinocytic activity value of 3.34 at the concentration of 2.5 mg/mL compared with the control, which had a value of 1.5 at the same concentration. Even when the concentration of the aqueous ethanol extract (70%) was 1.25 mg/mL, it had a significant effect with a value of 2.79, so these results suggested that this extract could activate the pinocytosis of macrophages. Moreover, the second extract, aqueous methanol (90%), also had a significant effect with a value of 3.12 at the concentration of 10 mg/mL, and the other two extracts, n-hexane and ethyl acetate, showed a mild effect in this regard (Figure 8b).

### 3.6. Apoptotic Activity of Carthamus oxyacantha Extracts against the T47D Cell Line

The apoptotic activity assay suggested that the levels of caspase-3 were elevated, suggesting the facilitation of programmed cell death and the inhibition of cellular proliferation (Figure 9). The results indicate that the ethyl acetate extracts exhibited the most significant activity, with a 3.29-fold increase in the levels of caspase-3 and *p* < 0.0001 compared with the negative control at concentrations of 0.14 mg/mL. The n-hexane and aqueous ethanol (70%) extracts demonstrated comparable levels of significant activity, with values of 1.91 and 1.94 observed at concentrations of 0.50 mg/mL and 4 mg/mL, respectively. The *p*-values associated with these observations were 0.0005 and 0.0004, respectively, compared with the negative control, as shown in the graph (Figure 9).

### 3.7. In Vivo Analysis of Carthamus oxyacantha M.Bieb Extracts

The pilot study on a small group of mice was conducted to determine the dosage ranges for the actual LD50. Six female mice (6 weeks, 20–23 g) were injected intraperitoneally (IP) with 0.8 g/kg of plant extract. They were monitored for 24 h for any signs of mortality. The next day’s dose was adjusted by increasing it 1.5-fold if it was well tolerated or decreasing it 75-fold if it was toxic (a dose range of 1200–1800 mg/kg was established for the actual LD50 (median lethal dose). Subsequently, during the phase-two trial, the LD_50_ was determined to be 1820 mg/kg utilizing the Karber method. Notably, 182 mg/kg of each extract was administered during our study, ten times lower than the toxic dose [22].

The EMT6/p cells were used to inoculate tumors subcutaneously into Balb/C mice. After seven days, the female Balb/C mice were injected intraperitoneally with 183.2 mg/kg of the two extracts, n-hexane and ethyl acetate (Figure 10). After waiting 19 days, they were sacrificed while continuing to weigh the tumors until day 20. The size of the tumors was determined, and tumor volume was calculated using a formula (Table 4). The size of tumors injected with n-hexane and ethyl acetate extracts was significantly reduced (*p* < 0.001) compared with the negative control group. Both n-hexane and ethyl acetate extracts had considerable antitumor activity, as measured by the size reduction of the tumors (−49.981% and −51.028%, respectively) compared with the control group (+8.0484%) (Table 4). Two instances of mouse death were reported for the control group, and one mouse showed no tumor after 14 days. The percentage of mice in treatment groups without detectable tumors was 37.5%. However, the mice exhibited typical behavior (the mice behaved in a manner that is considered normal and healthy for their species) without any adverse effects.

### 3.8. The Toxicity of the Extract Using AST, ALT, and Creatinine Tests

The study measured serum levels of ALT and AST in animals treated with *Carthamus oxycantha* extracts of n-hexane and ethyl acetate extracts. The study revealed that treated tumor-bearing mice exhibited elevated levels of AST, with a recorded value of 242.36 IU/L along with a *p*-value of <0.0001 for n-hexane and 449.25 IU/L with a *p*-value of 0.0256 for ethyl acetate, compared with the negative control mice with a recorded AST level of 176.60 IU/L. Despite this, the plasma ALT levels were insignificant for both extracts, with the n-hexane extract and ethyl acetate extract exhibiting values of 54.83 IU/L and 43.43 IU/L, respectively, in contrast to the negative control with a value of 32.53 IU/L. Nonetheless, these values fall within the normal range compared with healthy mice [25] (Figure 11). At the same time, the creatinine levels were also determined for mice treated with ethyl acetate and n-hexane extracts, and they were found to be within the normal range (Table 5).

## 4. Discussion

The escalating incidence and mortality rates of cancer have led to an elevated awareness of the significance of adjuvant and alternative cancer therapies [26]. The utilization of medicinal plants as a supplementary treatment alongside chemotherapy and radiotherapy has been observed to reduce the development of drug resistance and adverse effects of chemotherapy [27].

Phytochemicals and natural products have long been accepted as an origin of chemical diversity, providing the base for discovering promising new anticancer agents that may possess high efficiency and low toxicity. The genus *Carthamus* has been acknowledged as a crucial source of conventional medicine due to its immune-enhancing, antioxidant, and anticancer properties [28]. There are limited data associated with the biological investigation of *Carthamus oxyacantha*.

To the best of our knowledge, the present study is the first to evaluate *Carthamus oxyacantha* through immunomodulatory and antiproliferative evaluation in vitro and in vivo using several extracts.

The extracts of *Carthamus oxyacantha* exhibited significant inhibitory effects on cancer cells in a dosage-dependent manner. The findings indicate that the n-hexane and ethyl acetate extracts showed a significant inhibitory effect on the cellular growth of all cancer cell lines tested, as displayed in the results (Table 2). The n-hexane extract was the most potent extract against T47D and Caco-2 cell lines and had IC_50_ values of 0.067 mg/mL and 0.067 mg/mL, respectively. In comparison, ethyl acetate extract was active against T47D and MDA-MB231, and IC_50_ values were 0.0179 mg/mL and 0.03 mg/mL, respectively. On the other hand, the aqueous ethanol extract (70%) exhibited moderate activity against all cancer cell lines, while the aqueous methanol extract (90%) lacked a significant effect against all cell lines.

Most anticancer medications used nowadays in chemotherapy are cytotoxic to normal cells, leading to unwanted side effects. The current study provides a significant bearing in the search for compounds that can reduce the harmful side effects of anticancer drugs, as n-hexane and ethyl acetate extracts showed significant cyto-selective effect indicated by higher selectivity index values against T47D and Caco-2 for n-hexane extract and T47D and MDA-MB231 for ethyl acetate extract.

To further investigate the mechanism by which *Carthamus oxyacantha* extracts exert their effects, the induction of apoptosis was assessed using a caspase-3 activity assay. Apoptosis restraining is known to be one of the ways by which cancer cells secure proliferation and survival [29]. Therefore, apoptosis induction has been suggested as an efficient mechanism to counter cancer cell proliferation. The results revealed that the ethyl acetate extract of *Carthamus oxyacantha* had the highest impact. Caspase-3 activity was enhanced by 3.29-fold compared with the negative control. Other solvent extracts exhibited moderate to mild apoptotic effects.

Since *Carthamus oxycantha* extract exhibited a significant antiproliferative effect, there was a further investigation in a mouse model implanted with breast cancer. This revealed a reduction in tumor size accompanied by an increased cure percentage. The observed apoptosis induction and the decrease in tumor size can potentially be related to the cytotoxic activity of the phytoactive compounds, which significantly impact the tumor cells. The presence of phytochemicals, including flavonoids, phenolics, iridoids, and alkaloids, could explain these activities [30].

The proper functioning of the human immune system is vital for the organism’s survival against toxic, infectious, and oncogenic agents. The lymphocyte is the central adaptive immune system cell, and the macrophage is one of the most important innate immune cells in the body. Therefore, *Carthamus oxycantha* plant extracts induced the proliferation of splenic lymphocytes as well as the induction of phagocytosis and pinocytosis [31].

Our results showed increased splenic lymphocyte count following the administration of *Carthamus oxycantha* extracts, suggesting an immune-stimulatory impact on the acquired immune system [32]. The results indicated that *Carthamus oxycantha* extracts exhibit a varying degree of increased proliferation in both T and B lymphocytes. Plant products have previously been shown to mediate immuno-stimulatory properties [33].

Generally, several phytochemicals are studied for their immunomodulating and anticancer properties. Immunomodulators can fight cancer due to their anti-inflammatory, antioxidant, apoptosis-inducing, anti-angiogenesis, and anti-metastasis effects, and their ability to elicit and activate humoral and cell-mediated immune responses against the tumor that facilitate the recognition and destruction of the already existing tumor. Also, they can be used as prophylaxis against the initiation of cancer, in addition to inhibiting tumor growth and proliferation. Various phytochemical compounds have been studied and reported for their anticancer and immunomodulators [34]. Flavonoids were reported to possess little or no cytotoxic effect on healthy cells while being cytotoxic against various human cancer cells [35]. Several suggestions have been proposed to explain flavonoids’ anti-carcinogenic activities like induction of cell cycle arrest and apoptosis, mitigation of oxidative damage, carcinogen inactivation, inhibition of cell growth and differentiation, reduction of vascular proliferation, diminishing of tumor angiogenesis, and restriction of metastasis [35,36,37].

Conversely, flavonoids can promote specific immunomodulatory effects that are fundamental to the control and treatment of different types of cancers. For example, quercetin is a chemopreventive and anti-genotoxic agent. It is capable of stimulating the immune system by increasing the proliferation of NK cells [38,39].

Iridoids possess antitumor activity towards various cancer cell lines, controlling the signal transduction molecules involved in the proliferation and death of cancer cells, which can help in the therapeutic improvement of cancer [40].

On another hand, phenolic acids have been found to possess potent anticancer abilities in numerous in vitro and in vivo studies. Moreover, the therapeutic activities of phenolic acids are fortified by their role as epigenetic regulators and guardians against adverse events or resistance associated with conventional anticancer therapy [36]. Data suggest that phenolic acids can modulate the immune response and various transcription factors, reduce cytokine secretion, and enhance the innate immune system [41,42].

The alkaloid piperine has been reported to elicit anticancer effects through diverse mechanisms of action, which involve the modulation of cell cycle progression, antioxidant activities, and anti-apoptotic actions [43]. Also, piperine has a strong affinity with CD4 and CD8 receptors, suggesting its potential as a potent immunomodulator [44].

Both ALT and AST enzymes are markers that have been used to assess the presence of liver damage. Furthermore, creatinine serum is used as a marker for assessing renal function (neem leaf (*Azadirachta indica* A. Juss) ethanolic extract on the liver and kidney function of rats). The treated group in this study exhibited normal-range creatinine and ALT levels [25]. These findings suggest that all treatments had an acceptable safety profile. These findings may be justified because the doses employed in this investigation were calculated using the Karber method and based on LD_50_ estimation, with no adverse consequences [45]. Conversely, the AST levels recorded in the present study were relatively elevated in the group receiving treatment compared with the control group. Currently, there is no definitive explanation for the elevation of AST. However, previous studies using *Carthamus tinctorius* have reported a substantial hepatoprotective effect and maintaining the structural and functional integrity of hepatocytes [46,47].

Plants and herbs include a wide range of complex chemical ingredients that affect the organism as a whole or individual organs and systems. Some chemical elements are mild and safe even in high dosages, but others are more potent or potentially dangerous in high doses or when taken continually [48].

Phytochemicals and plants’ secondary metabolites are significant contributors to various aspects of health, including medicine, nutrition, and disease prevention [49]. The present study involved the phytochemical analysis of two extracts that exhibited the greatest activity level among the four. The antiproliferative activity of *Carthamus oxycantha* extracts is mediated by its bioactive constituents. Plants are a source of compounds for the development of novel anticancer medicine. Various compounds were identified, including flavonoids, alkaloids, terpenes, and phenolics. It is expected that these compounds act synergistically to produce the observed outcomes (Figure 12) [50].

*Carthamus oxyacantha* contains numerous fascinating components that have been extensively researched for medicinal purposes. The diverse array of phytochemicals in this plant, even in small quantities, enables us to comprehend better its therapeutic properties like anticancer and immune-enhancing properties.

Since the cytotoxic activity of the plant extracts in cell lines has been demonstrated, it is necessary to carry out a bioassay-guided study to isolate and characterize the bioactive compounds responsible for this effect and to evaluate their mechanism of action further to understand the medicinal impact of this plant against cancer.

## 5. Conclusions

*Carthamus oxyacantha* M.Bieb is a potential source of numerous active phytochemicals, which may render it a promising candidate for targeting breast and colon cancer. This study also indicated that *Carthamus oxyacantha* has immunomodulatory properties due to its ability to stimulate splenic lymphocyte proliferation and the induction of phagocytosis and pinocytosis. Furthermore, the in vivo testing also yielded promising results regarding the average reduction in tumor size, which can be considered a reliable depiction of the preventive impact of *Carthamus oxyacantha*. Although the exact mechanisms are not fully understood, it has been established that the plant induces apoptosis and modulates the immune system. Finally, this fascinating plant requires further investigation to expand the scope of research and establish additional knowledge for its potential use in cancer treatment. Also, using more solvent extract may help to extract more active phytochemicals.

## Figures and Tables

**Figure 1 plants-13-00042-f001:**
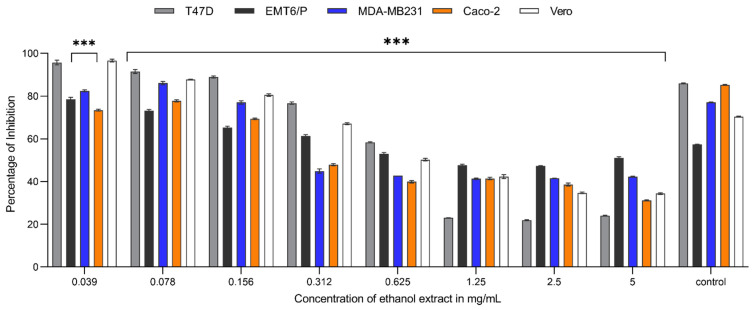
Anti-proliferative activity of different concentrations (0.0390625 mg/mL to 5 mg/mL) of aqueous ethanol (70%) extract of *Carthamus oxyacantha* on cancer cell lines (T47D, EMT6/P, MDA-MB231, Caco-2) and Vero normal cell line. Results represented as mean ± SD (*n* = 3). Significance of results represented by *p* of <0.001 tagged by (***) asterisks. All concentrations are compared with mean values of negative control.

**Figure 2 plants-13-00042-f002:**
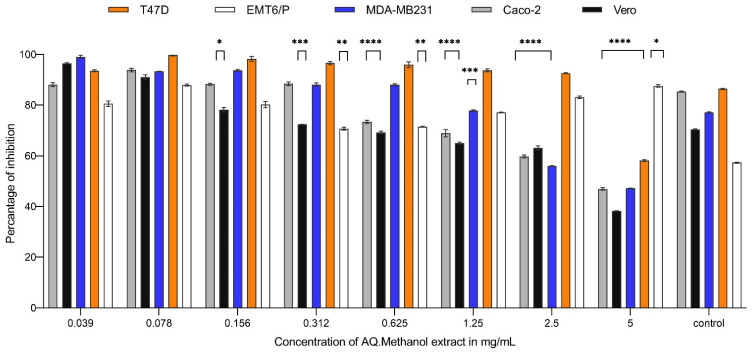
Anti-proliferative activity of different concentrations (0.0390625 mg/mL to 5 mg/mL) of AQ. methanol extract (90%) of *Carthamus oxyacantha* on cancer cell lines (T47D, EMT6/P, MDA-MB231, Caco-2) and Vero normal cell line. Results represented as mean ± SD (*n* = 3). Significance is represented by (*) *p*-value < 0.05, (**) *p*-value < 0.001, (***) *p*-value < 0.001 and (****) *p*-value < 0.0001 significance level. All concentrations are compared with mean values of negative control.

**Figure 3 plants-13-00042-f003:**
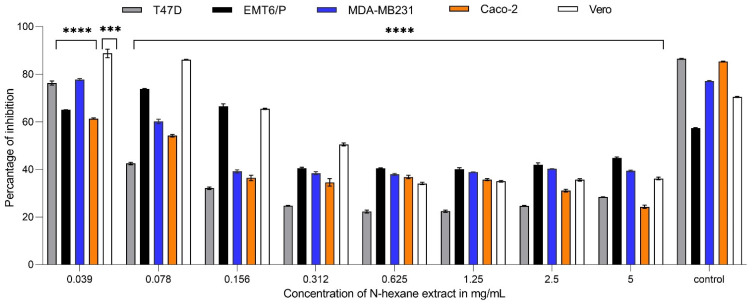
Anti-proliferative activity of different concentrations (0.0390625 mg/mL to 5 mg/mL) of n-hexane extract of *Carthamus oxyacantha* on cancer cell lines (T47D, EMT6/P, MDA-MB231, Caco-2) and on Vero normal cell line. Results represented as mean ± SD (*n* = 3). Significance is represented by (***) *p*-value < 0.001 and (****) *p*-value < 0.0001 significance level. All concentrations are compared with mean values of negative control.

**Figure 4 plants-13-00042-f004:**
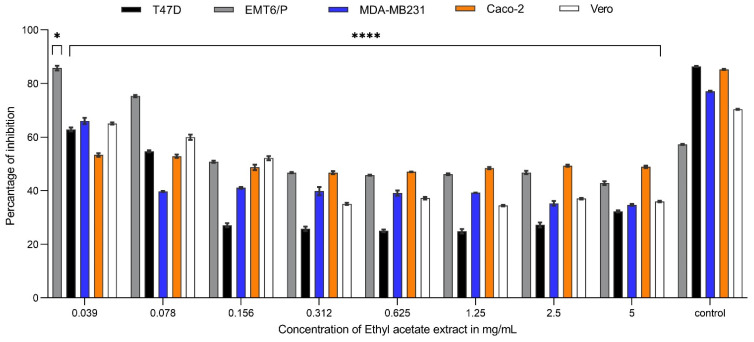
Anti-proliferative activity of different concentrations (0.0390625 mg/mL to 5 mg/mL) of ethyl acetate extract of *Carthamus oxyacantha* on cancer cell lines (T47D, EMT6/P, MDA-MB231, Caco-2) and on Vero normal cell line. Results represented as mean ± SD (*n* = 3). Significance is represented by (*) *p*-value < 0.05, and (****) *p*-value < 0.0001 significance level. All concentrations are compared with mean values of negative control.

**Figure 5 plants-13-00042-f005:**
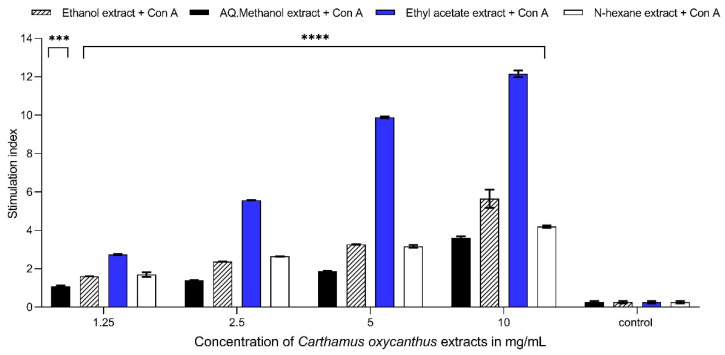
Lymphocyte proliferation assay using Con A as an activator for different *Carthamus oxycanthus* extracts (aqueous ethanol, aqueous methanol, ethyl acetate, n-hexane) using concentrations between 1.25 mg/mL and 10 mg/mL. Results represented as mean ± SD (*n* = 3). Significance is represented by (***) *p*-value < 0.001 and (****) *p*-value < 0.0001 significance level. All concentrations are compared with mean values of negative control.

**Figure 6 plants-13-00042-f006:**
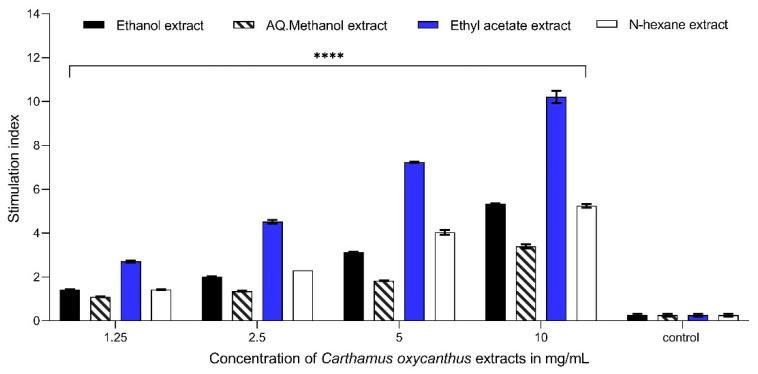
Lymphocyte proliferation assay using LPS as an activator for different *Carthamus oxycanthus* extracts (aqueous ethanol, aqueous methanol, ethyl acetate, n-hexane) using concentrations between 1.25 mg/mL and 10 mg/mL. Results represented as mean ± SD (*n* = 3). Significance is represented by *p* < 0.0001 tagged by (****) asterisks. All concentrations are compared with mean values of negative control.

**Figure 7 plants-13-00042-f007:**
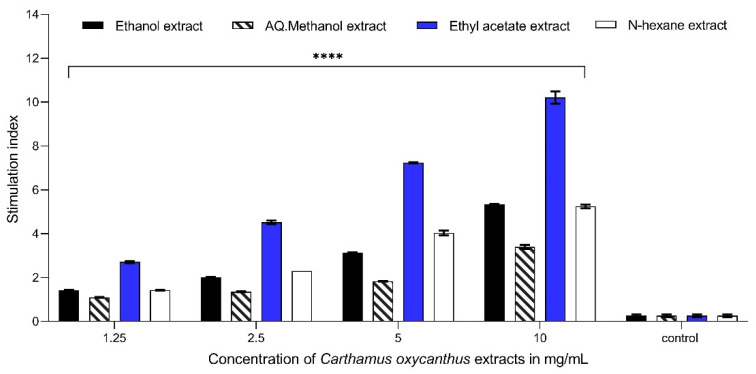
Lymphocyte proliferation assay without the presence of Con A or LPS for different *Carthamus oxycanthus* extracts (aqueous ethanol, aqueous methanol, ethyl acetate, n-hexane) using concentrations between 1.25 mg/mL and 10 mg/mL. Results represented as mean ± SD (*n* = 3). Significance is represented by *p* < 0.0001 tagged by (****) asterisks. All concentrations are compared with mean values of negative control.

**Figure 8 plants-13-00042-f008:**
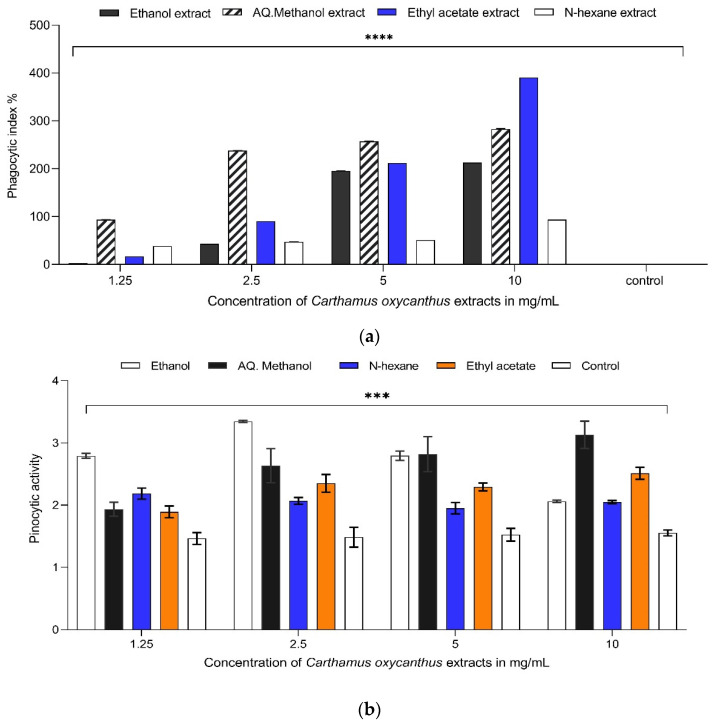
(**a**) The effects of *Carthamus oxyacantha* extracts on phagocytosis. The percentage of the phagocytic index for peritoneal macrophage with different concentrations (1.25 mg/mL to 10 mg/mL) of different *Carthamus oxyacantha* extracts (aqueous ethanol extract (70%), aqueous methanol (90%), n-hexane, ethyl acetate). Results represented as mean ± SD (*n* = 3). Significance is represented by *p* ≤ 0.0001 tagged by (****). All concentrations are compared with mean values of negative control. (**b**) The effects of *Carthamus oxyacantha* extracts on pinocytosis. The pinocytotic activity of peritoneal macrophage with different concentrations (1.25 mg/mL to 10 mg/mL) of *Carthamus oxyacantha* extracts (aqueous ethanol extract (70%), aqueous methanol (90%), n-hexane, ethyl acetate). Results represented as mean ± SD (*n* = 3). Significance is represented by *p* < 0.001 tagged by (***). All concentrations are compared with mean values of negative control.

**Figure 9 plants-13-00042-f009:**
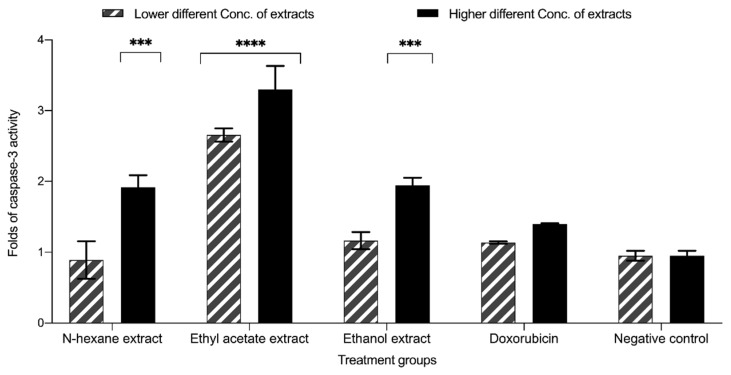
The effect of IC_50_ concentration of *C. oxycantha* extracts on caspase-3 activity in the T47D cell line. The concentrations of the extracts: n-hexane (0.5 mg/mL), ethyl acetate (0.014 mg/mL), and aqueous ethanol (70%) (4 mg/mL). Results represented as mean ± SD (*n* = 2). Significance is represented by (***) *p*-value < 0.001 and (****) *p*-value < 0.0001 significance level. The results were calculated by dividing the reading for each treatment by the reading of the negative control.

**Figure 10 plants-13-00042-f010:**
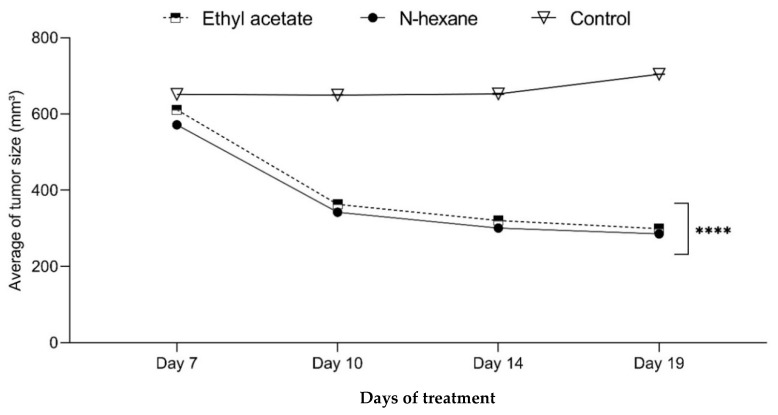
The effects of two different extracts (n-hexane and ethyl acetate) on the average tumor size of Balb/C mice bearing the EMT6/P tumor cell line. Results represented as mean ± SD (*n* = 3). Significance is represented by *p* of < 0.0001 tagged by (****) asterisks. All concentrations are compared with mean values of negative control.

**Figure 11 plants-13-00042-f011:**
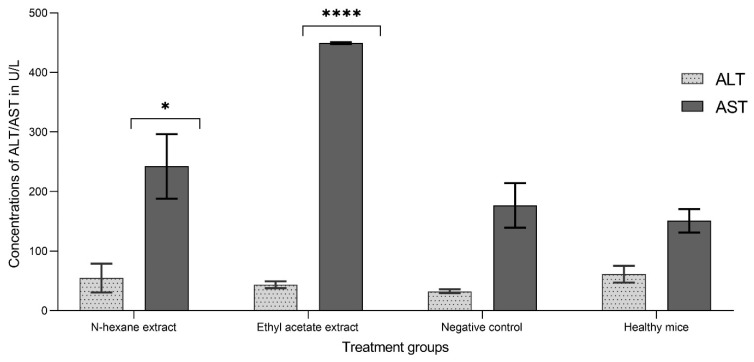
The effects of two different extracts (n-hexane and ethyl acetate) on serum ALT and AST in Balb/C mice bearing the EMT6/P tumor cell line. Results represented as mean ± SD (*n* = 3). Ethyl acetate extract showed significance in AST concentration represented by *p* < 0.0001 tagged by (****) asterisks, and n-hexane extract showed significance in AST concentration represented by *p* of <0.0146 tagged by (*) asterisks. All concentrations are compared with the mean values of the negative control.

**Figure 12 plants-13-00042-f012:**
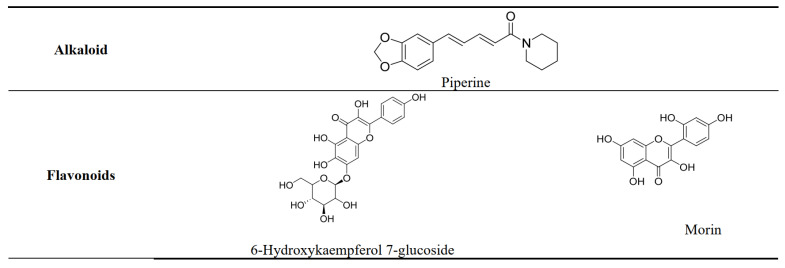
The structure of phytochemical compounds with potential anticancer and immunomodulatory activities identified in the ethyl acetate and n-hexane extracts of *Carthamus oxyacantha*.

**Table 1 plants-13-00042-t001:** The IC_50_ of aqueous ethanol, aqueous methanol, n-hexane, and ethyl acetate extracts of *Carthamus oxyacantha* using various cell lines compared with the normal Vero cell line and the anticancer drug doxorubicin.

Cell Line	IC_50_ of Aqueous Ethanol (70%) (mg/mL)	IC_50_ of Aqueous Methanol (90%) (mg/mL)	IC_50_ of n-Hexane (mg/mL)	IC_50_ of Ethyl Acetate (mg/mL)	IC_50_ of Doxorubicin (μg/mL)
T47D	0.641 ± 0.025	>5	0.067 ± 0.005	0.0179 ± 0.012	7.01 ± 0.4
EMT6/P	0.816 ± 0.049	>5	0.398 ± 0.012	0.497 ± 0.016	0.57 ± 0.4
MDA-MB231	0.485 ± 0.037	2.98 ± 0.008	0.14 ± 0.01	0.03 ± 0.05	15.56 ± 0.5
Caco-2	0.497 ± 0.025	>5	0.067 ± 0.025	0.214 ± 0.037	0.47 ± 0.31
Vero	0.753 ± 0.013	4.7 ± 0.035	0.46 ± 0.012	0.251 ± 0.012	>200

**Table 2 plants-13-00042-t002:** The selectivity index of aqueous ethanol, aqueous methanol, n-hexane, and ethyl acetate extracts against T47D, EMT6/P, MDA, and Caco-2 cell lines.

Cell Line	SI of *Carthamus oxyacantha* M.bieb Extracts			
	Aqueous Ethanol (70%)	Aqueous Methanol (90%)	n-Hexane	Ethyl Acetate
T47D	1.17	-	6.86	14
EMT6/P	0.92	-	1.15	0.50
MDA-MB231	1.41	-	1.74	8.36
CACO-2	1.51	-	6.68	1.17

**Table 3 plants-13-00042-t003:** Phytochemical compounds detected in ethyl acetate and n-hexane extracts of *Carthamus oxyacantha* using LC-MS.

No.	Compound	Molecular Formula	*m*/*z*	Retention Time	Ethyl Acetate	n-Hexane
1	Shanzhiside methyl ester	C_17_H_26_O_11_	429.1369	5.98	D *	ND *
2	Cyanidine	C_15_H_11_O_6_^+^	287.11	9.54	D	ND
3	4-hydroxybenzoic acid	C_7_H_6_O_3_	139.0391	10.18	D	ND
4	Syringic acid	C_9_H_10_O_5_	199.0603	10.75	D	ND
5	Harpagid	C_15_H_24_O_10_	387.1632	10.94	D	ND
6	Quercetin 3,4’, di-O-glucoside	C_27_H_30_O_17_	627.1551	11.06	D	ND
7	Isorhamnetin 3-O-glucoside	C_22_H_22_O_12_	479.1195	11.4	D	ND
8	1,3,5trihydroxy benzene	C_6_H_4_O_3_	127.0393	11.44	D	D
9	1-Caffeoylquinic acid	C_16_H_18_O_9_	355.1026	11.55	D	D
10	6-Hydroxykaempferol 7-glucoside	C_21_H_20_O_12_	465.1	12.61	D	D
11	Ferulic acid	C_10_H_10_O_4_	195.0660	13.57	D	D
12	Quercetin-3-O-glucose-6-acetate	C_23_H_22_O_13_	507.1105	13.59	ND	D
13	2-(Hydroxymethyl)benzoic acid	C_8_H_8_O_3_	153.0549	13.67	D	D
14	Syringaldehyde	C_9_H_10_O_4_	183.0656	13.7	D	D
15	Coumarin	C_9_H_6_O_2_	147.0445	14.52	D	ND
16	Luteolin	C_15_H_10_O_6_	287.0551	14.80	D	ND
17	Morin	C_15_H_10_O_7_	303.0492	14.88	D	ND
18	Apigenin	C_15_H_10_O_5_	271.0607	15.71	D	ND
19	Piperine	C_17_H_19_NO_3_	286.1422	18.49	D	D
20	Dodecanedioic acid	C_12_H_22_O_4_	231.2097	26.05	ND	D

* D: detected, ND: not detected.

**Table 4 plants-13-00042-t004:** The effect of *Carthamus oxyacantha* on tumor size and weight in mice (*n* = 8) (mm^3^: cubic millimeters).

Group	Av. Initial Tumor Size (mm^3^) ± SEM	Av. Final Tumor Size (mm^3^) ± SEM	% Change in Tumor Size	% of Mice with No Detectable Tumor	Av. Tumor Weight [24]
Control *n* = 8	651.99 ± 1.901	704.47 ± 1.19	8.048	12.5	846
n-hexane *n* = 8	571.81 ± 0.64	286.01 ± 0.247	−49.98	37.5	276.4
Ethyl acetate *n* = 8	611.49 ± 0.340	299.46 ± 0.656	−51.02	37.5	202

**Table 5 plants-13-00042-t005:** The serum creatinine levels in (µmol/L) for the treatment group, the negative control, and the healthy mice.

Group	Creatinine (µmol/L) ± SEM
Healthy mice	0.18 ± 0.005
Negative control	0.14 ± 0.014
Ethyl acetate extract	0.16 ± 0.017
n-hexane extract	0.11 ± 0.012

## Data Availability

Data are contained within the article.

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
