# Peer review of "Anticancer, Immunomodulatory, and Phytochemical Screening of Carthamus oxyacantha M.Bieb Growing in the North of Iraq"

_plants, 2023, doi:10.3390/plants13010042_

Round 1

Reviewer 1 Report

Comments and Suggestions for Authors

Line 108, 121, 129, etc...delete the point in the top of each line

Why the authors used organic solvant only, a comparison with aqueous solvant is recommended 

Why the ethanol is used 90 and 70%? A better study will be 95% and 75%...

Some references need their abbreviation 

Line 279 the % of inhibition caused by the 4 solvants stay so closely   why choosing this 4 solvants ? Indeed you can add water and chloroform for a better results 

Reference 5 in italic  must be corrected 

Do you try a separation method by chromatography on silica gel for example and study the biological effect of each fraction or pure compounds?

Author Response

Thank you for your comments. Please see attached our response. All changes were highlighted in yellow color

Reviewer 2 Report

Comments and Suggestions for Authors

This study investigated the antiproliferation and immunomodulatory activity of Carthamus oxyacantha root against four kinds of human cancer cells. The prophylactic effect of Carthamus oxyacantha on Balb/C female mice implanted with the EMT6/p breast cancer cell line was studied. The results showed that the extracts of Carthamus oxyacantha could upregulate the immune system in vitro and in vivo, thus inhibiting the proliferation of tumors. In my opinion, this is an experimental complete manuscript, but there are some problems.

A brief introduction to the selected cell line is recommended, and the order of the tables in the diagram is chaotic. Therefore, the results of 3.1 and 3.2 need to be modified.

The significance comparison in the figure, I think, it should be the same treatment of different concentrations, or the same concentration of different treatments, rather than all mixed together for multiple comparative analysis.

In addition, the description of the method is divided into too many chapters, it is recommended to merge 2.9 and 2.10, 2.11 and 2.12,2.13.

Finally, the discussion is too verbose, and it is suggested to revise it.

Author Response

(The authors gave the same response as above.)

Reviewer 3 Report

Comments and Suggestions for Authors

This is a report of the authors attempts “to discover the potential immunomodulatory and cytotoxicity of different extracts of Carthamus oxycantha roots.” In brief, the authors applied preparations of C. oxycanth exposed to ethanol (90%, 70%), ethyl acetate, and n-hexane to five epithelial cell lines (T47D (Human CaBreast), MDA-MB231 (Human CaBreast), Caco-2 (Human CaColon), EMT6/P (Mouse CaBreast), and VERO Monkey, kidney), “…to uncover the therapeutic capability of Carthamus oxyacantha and its potential applications in cancer medicine by conducting an extensive investigation into its various characteristics and attributes.and its potential applications in cancer medicine by conducting an extensive investigation into its various characteristics and attributes

As a general observation, this reviewer finds the experimental work is inadequately described as to allow others ‘practiced in the art’ to reproduce the protocols and to acquire equivalent, or similar, results. 

Ln 65. This reviewer suggests that the paragraph starting “Cancer immunotherapy…” is not relevant to the subject of this study. While I do recognise some natural products possess immunomodulatory activity, I suggest, however, that most readers will misinterpret your reference to immunotherapy to mean immune cell-based therapy (e.g. CAR-T, NK, autologous therapies). In my view immunomodulation and immunotherapy are distinct, inequivalent modalities.

Ln 112. In the interests of reproducibility, please specify how… “plant samples were cleaned thoroughly”?

Ln 114. In the interests of reproducibility, please specify how… “dried roots were crushed”?

Ln 116. In the interests of reproducibility, please specify the formulation for “70% ethanol, n-hexane, aqueous methanol (90:10), and ethyl acetate”? 

Ln 117. In the interests of reproducibility, please elaborate the meaning of “daily stirring”? In the interests of reproducibility, please specify how “residue” was “removed”?

Ln 125. In the interests of reproducibility, please specify how “samples were washed”?

Ln 131. Please identify the source(s) of T47D, MDA-MB231, Caco-2, EMT6/P, and VERO cell lines used in this study? Please also specify how these cells lines were authenticated and when they were last tested for mycoplasma contamination?

Lns 133 – 135. From the descriptions, should the reader presume all cells lines were cultivated without serum?

Ln 142. In the interests of reproducibility, please specify the source of “96-well cell culture plate”?

Ln 143. In the interests of reproducibility, please specify the final concentrations of test compounds “…added to the concentration of 5 – 0.078 mg/mL”, and the solute (e.g. DMSO, ethanol, acetone, acid, water)? Please also specify the composition of control samples (e.g. solute alone, doxorubicin)? 

The sentence starting “Since doxorubicin…” is incomplete. Please revise. 

Ln 145. Please identify the origin of the “MTT assay” protocol and source of reagents? A reference and supplier are required.

Ln 148. Please specify how “SPSS was used to determine the IC50 values”? What is SPSS?

Ln151. For consistency, please revise “IC50” to “IC50” – or IC50” to IC50”?

What is the rationale for “the selectivity index (SI) was based on the ratio of IC50 of Vero cells 151 to human cancer cells”?

Ln 153. What is “a certain threshold”? (sic)

Ln 154. What is “the 𝑛𝑜𝑟𝑚𝑎𝑙 𝑐𝑒𝑙𝑙 𝑙𝑖𝑛𝑒”? (sic)

Ln 159. The authors specify “three treatments”; however, five treatments are reported: “ethanol extract…, n-hexane extract…, ethyl acetate extract…, doxorubicin hydrochloride…, a negative control“(unspecified). Pleas revise.

What was the rationale for selecting the reported concentrations” (ethanol extract) 4 mg/mL, (n-hexane extract) 0.5 mg/mL, (ethyl acetate) 0.014mg/mL, and (doxorubicin hydrochloride) 250 nM” for the test samples? Are these concentrations equimolar? Are these concentrations based on existing practices? Are these concentrations based on each compound’s IC50?

Ln 162. In the interests of reproducibility, please specify under what conditions were “cells… incubated for 48 hours”?

Ln 163. Sigma-Aldrich sells several Caspase 3 Assay kits. In the interests of reproducibility, please specify which Caspase 3 Assay product was used?

Ln 168. Please provide specifications for the” 0.45 μm membrane filter”? What is the evidence that the filter material did not retain any target phytochemicals and thereby affect the final concentration of bioactive reagents?

Ln 180. Please identify the gender distribution of “Balb/C mice”? 

Ln 182. For reproducibility, please specify what is “the standard ventilation system”? Light cycle (unreported) may also be an important study parameter.

Ln 184. For reproducibility, please describe the “aseptic extraction of the splenocytes”?

Ln 185. For reproducibility, please identify how “…the cells were processed in the tissue grinder” (sic)? What is “the tissue grinder”?

Pg 186. Reporting “a speed of 2000 rpm” is meaningless, without also reporting the diameter of the rotor, or the model of centrifuge!

Ln 187. Please identify the “RBC lysis buffer” (sic)? How were “splenocytes… counted”?

Ln 191. For reproducibility, please identify the concentrations used for “Con A and LPS”?

Ln 192. As currently reported, the “Lymphocyte proliferation assay” protocol in which “Splenocyte suspension was prepared… in RPMI1640” (sic), will not yield meaningful results. Freshly isolated splenocytes will not survive in RPMI 1640 without the addition of supplements – i.e. L-glutamine, serum! Please revise. FYI, the presence of serum is also critical for cell recognition of LPS!

Ln 195. For reproducibility, please identify “MTT solution”?

Ln 196. How was “absorbance… measured with an ELISA microplate reader” (sic)? 

Ln 197. How was “percentage of proliferation” determined?

Ln 201. Please identify the source of “Brewer's thioglycollate medium”?

Ln 202. For reproducibility, please describe how were “peritoneal macrophages” collected?

Ln 208. Please identify the source and composition of “yeast suspension” (sic)?

Ln 217. The “neutral red technique” reported by “Boothapandi et al.” is derivative. Please cited the original publication: Płytycz B, et al. Folia Biol (Krakow). 1992. 40(1-2):3-9. PMID: 1451835.

Ln 232. What is the rationale for “the… dose was adjusted by increasing it by 1.5-fold if it was well-tolerated or decreasing it by 75-fold if it was toxic” (sic)? This is not consistent with ISO10993.

Ln 254. Please identify the source of “Serum samples” (sic)?

Ln 260. For reproducibility, please identify “Working reagents” (sic)?

Ln 303. What is the meaning of “decent activity” (sic)? 

Lns 319, 324, 335, 349. Figures 1, 2, 3, and 4, are Titled: “Anti-proliferative activity…”, yet each figure reports “Percentage of survival” measures. “Survival” is not a measure of “Anti-proliferative activity”. Please revise.

The ordinate in Figures 1, 2, 3, and 4, are labelled “Percentage of survival”; however, is annotated “0.0 to 1.0”. This is incorrect. It should be annotated ‘0% to 100%’. As currently reported, animal survival is exceedingly (unacceptably) low (i.e. <1.0%)! Please revise.

Ln 386. Table 5. This reviewer is confused. How does LC-MS… “F: found” or “NF: not found” reagents/species? Surely LC-MS ‘detects, resolves, identifies’? LC-MS does not ‘find’! Please revise.

Ln 393. Please determine “highest efficacy” …of what?

Ln 395. Please identify “mitogens”? This reader has not identified the addition of any mitogens in the Materials and Methods? I am confident that the average reader would not understand Con A and LPS are mitogens in this assay!

Ln 397. This reviewer is not familiar with a “stimulation index”. Would the average reader understand this measure? 

Ln 422. Please identify the data to substantiate the claim: “assay results indicate that the ethyl acetate extract, at a concentration of 10 mg/mL, demonstrated the highest level of peritoneal phagocytic activity”? A reference to the data is required.

Ln 423. Would the average reader understand “Phagocytic index”? 

Ln 425. I am concerned; how does “methanol extract… exhibit… phagocytic activity”? (sic) My understanding is that phagocytosis is a property of mammalian cells. Phagocytosis is not a property of solvents (i.e. “methanol extract(s))”! Please revise.

Ln 430. I can accept that most readers appreciate that macrophages are professional phagocytic cells; however, will most readers also appreciate that macrophages are capable “pinocytotic activity”? Would the average reader understand that phagocytosis and pinocytosis are distinct biological activities with distinct mechanisms? 

Please identify the data that substantiates the claim “pinocytotic activity of the cells was enhanced.” (sic)

Ln 452. Please identify the evidence (data) that substantiates the claim “…apoptotic activity assay suggested that the levels of caspase-3 were elevated”? (sic) 

Ln 455. I am very skeptical that “a 3.29-fold increase in the levels of caspase-3” represents “p <0.0001” of statistical significance. From biological and physiological perspectives, this statement of significance is meaningless!

Ln 459, In common with Ln 455, the statement claiming statistical significance of the response of T47D exposed to n-hexane and ethanol extracts of C. oxyacantha, “0.0005 and 0.0004“, is meaningless overstatement.

Ln 461. This reader is confused.  The ordinate of Figure 9 is labelled “Folds of caspase-3 activity“(sic), however, the Figure’s Title refers to “caspase-3 expression”. ‘Expression’ is not the same as ‘activity’. It is not stated in the figure legend how these assay data were obtained. Please revise. 

Ln 466. Incomplete sentence: “…a dose range of 1200-1800 mg/kg was established” …for what? This sentence requires a subject.

Ln 468. Please cite original source for “Karber method” (i.e. [22])

Ln 473. This reader does not. Understand the description: “After waiting 19 days, (animals were) sacrificed while continuing to weigh the tumors until day 20” (sic). Please revise.

Please describe. How were “size of the tumors… determined”? (sic)

Ln 481. What is “typical behaviour”? (sic)

Ln 512. It is not clear from where measures for “serum creatinine levels” are obtained? How does one achieve absorbance measures acquired at 500 nm to 5 decimal places of sensitivity? This is simply unachievable using (undocumented, unidentified) standard laboratory spectrophotometry instruments. These reported data are not believable.

Author Response

(The authors gave the same response as above.)

Reviewer 4 Report

Comments and Suggestions for Authors

The usage of LD50 is no more nowadays. The authors should use a toxicity study instead of LD50. 

Authors could have established the mechanism of action by doing Cell cycle analysis, nexin-FITC/PI. DNA fragmentation assay, etc

Author Response

Thank you very much for your comments.

We revised the manuscript  the conclusion was modified to show the importance of doing more assays to clearly understand the mechanisms of action and improve the toxicity assay.

Round 2

Reviewer 3 Report

Comments and Suggestions for Authors

This is a revised version of a report describing experiments aimed “to discover the potential immunomodulatory and cytotoxicity of different extracts of Carthamus oxycantha roots.” I reviewed the first version and found that (as described) was not reproducible, poorly validated, and overinterpreted. I did not support it publication in Plants.

The authors have accepted almost all my comments and have revised their manuscript accordingly. However, I find this revised version still includes too many unsubstantiated statements of opinion, lack of validation, lack of authentication, inappropriate statistics, lack of consistency and attention to detail (sloppiness), overreach, and overinterpretation. 

Ln 29. How do “extracts had considerable size reduction”? How do extracts change size? I suggest you mean ‘extracts effected/induced/caused…reductions in the size of tumours’? Please revise.

Ln 30. How is it possible to measure (wet?) tumour size accurately to 3 decimal places? I suggest that this precision exceeds the accuracy of standard laboratory measuring devices! To quote this level of accuracy is meaningless in this context. On the contrary, claims such as these undermine the credibility of the authors. Please revise.

Why is it ‘remarkable’, that “Carthamus oxyacantha extracts decreased the average weight of the tumor cells in vivo”?

Subjective opinion has no place in scientific communication. Please revise.

Ln 31. When reporting experimental evaluations of ‘solvent extracts’, is it inaccurate to claim, “The plant induced significant apoptotic activity…”? Please revise.

Ln 98. Please define the abbreviations: RPMI, MEM, DMEM, FBS, PBS, EDTA, DPPH.

Ln 122. It is not clear to me why the authors subjected C. oxycantha to “aqueous ethanol (70%), aqueous methanol (90%), ethyl acetate, and n-hexane for 14 days at room temperature (Ln 116), yet subjected C. oxycantha to “ethyl acetate and n-hexane” (Ln 125) for 4 hours? How can these protocols be considered comparable?

Ln 127. For consistency, please replace “Rotavapor.. dried out” (Ln 127) with “rotary evaporator and lyophilized” (Ln 119)? 

Ln 132. What is the evidence that the specified cells lines are authentic and free from cross-contamination? See doi: 10.1007/s11626-998-0040-y. Please provide evidence cell lines have been tested for cross-contamination, and for microbial contamination. www.atcc.org/resources/technical-documents/cell-line-authentication-test-recommendations.

Ln 139. For consistency, please replace T47d” with ‘T47D’?

Ln 149. Poor grammar! Please replace “done” with ‘assayed’?

Ln 151. For reproducibility, please identify “After incubation…” Under what conditions were cells “incubated”, prior to treatment “with various concentrations of C. oxyacantha extracts”? 

Ln 153. What is the rationale for “After 1 day or 2 days of incubation” (sic)? 

Ln 157. Incomplete sentence. “The IC50 values and the percentage of surviving cells” (sic) is lacking a verb!

Ln 190. “Inertsustan C18” should read ‘InertSustain C18’. Please identify the supplier (i.e. GL Sciences)

Ln 230. Brewer’s thioglycollate (sic) medium requires a reference: JAMA. 1940;115(8):598-600. doi:10.1001/jama.1940.72810340001009.

Ln 232. The protocol “cells were centrifuged at 2000 RPM” (sic) is meaningless without also reporting the centrifuge rotor’ diameter. It is preferred this be reported as RCF, to enable others to reproduce using alternative equipment.

Ln 235. Please define “NBT” with first use?

Ln 238. In the interest of reproducibility, please identify “yeast suspension”? 

Ln 313. For consistency, please replace “MDAMBA231cell” with ‘MDA-MB231 cell’?

Ln 320. Please revise “mg/m” (sic) to ‘mg/mL’?

Ln 341. While previous IC50 values are reported as X mg/mL, “IC50 values of Doxorubicin against the T47d (sic), EMT6/P, MDA, Caco-2, and Vero Cells lines” are reported without unit measures. Please revise.

Ln 342. For consistency, please replace “MDA” with ‘MDA-MB231’.

Ln 351. Fig 1. Please edit ordinate title: “Concentration of ethanol extract im mg/ml” (sic)? This should read ‘…extract in mg/mL’.

Ln 353. For consistency, please replace “MDAMB231” with ‘MDA-MB231’?

Ln 358. Fig 2. Please edit ordinate title: “Concentration of AQ.Methanol extract im mg/ml” (sic)? This should read ‘…extract in mg/mL’. Please define “AQ.Methanol”?

Ln 361. For consistency, please replace “MDAMB231” with ‘MDA-MB231’?

Ln 362. This reader does not comprehend the meaning of “Significance is represented by a (p of 0.0146) tagged by (*) asterisks and less all concentrations are compared to mean values of the negative control.” (sic). Please revise.

Ln 365. Fig 3. There is no Figure 3 included in the document provided to me!

Ln 367. For consistency, please replace “MDAMB231” with ‘MDA-MB231’?

Ln 368. This reader does not comprehend the meaning of “Significance is represented by a (p of 0.000) tagged by (***) asterisks, and lower concentrations are compared to mean values of negative control.” (sic) Please revise.

Ln 372. Please edit ordinate title: “Concentration of Ethyl acetate extract im mg/ml” (sic)? This should read ‘…extract in mg/mL’. 

Ln 374. For consistency, please replace “MDAMB231” with ‘MDA-MB231’?

Ln 375. This reader does not comprehend the meaning of “Significance is represented by a (p of 0.0146) tagged by (*) asterisks and less all concentrations are compared to mean negative control values.” (sic)

Ln 380. A unnamed bar chart is evident at line 380. It is without a legend. I interpret this might be a lay-out widow, possibly the missing Fig 3!

Please edit ordinate title: “Concentration of n-hexane extract im mg/ml” (sic)? This should read ‘…extract in mg/mL’.

Ln 280. For consistency, please replace “MDAMB231” with ‘MDA-MB231’?

Ln 382. For consistency, please replace T47d” with ‘T47D’?

Ln 383. For consistency, please replace “MDAMB231” with ‘MDA-MB231’?

Ln 386. Please avoid using abbreviations in Figure Titles; “The SI of…” (sic)

Ln 387. For consistency, please replace “MDAMB231” with ‘MDA-MB231’?

Ln 388. For consistency, please replace “MDAMB231” with ‘MDA-MB231’?

Ln 391. For consistency, please replace “MDA” with ‘MDA-MB231’?

Ln 397. For consistency, please replace “MDAMB231” with ‘MDA-MB231’?

Ln 413, Ln 414. These appear to be editing widows. They are blank and interrupt Table 3.

Ln 431. This reader does not comprehend the meaning of “Significance is represented by (p of 0.0146) tagged by (***) asterisk, and less all concentrations are compared to mean values of the negative.” (sic)

Ln 483. It is not obvious to me that elevated caspase-3 “suggest(s) the facilitation of programmed cell death and the inhibition of cellular proliferation” (sic)? Caspase-mediated cell death and cell proliferation are independent metabolic events. I suggest that the authors are confusing proliferation assay data with individual cell outcomes. Proliferation assays report population data, however. A weaker signal does not necessarily mean less proliferation, nor does it mean increased cell death. MTT assays measure mitochondrial metabolic activity – only! Detection of caspase 3 does not automatically imply the initiation of programmed cell death. For example, caspase 3 is also involved in motility and metastatic invasion.

Ln 499. Please identify the relevant in vivo data for “the pilot study” (sic)? 

Ln 500. Please identify the protocol for “the phase two trial” (sic)? The provided citation [22] is not for this pilot study, nor phase one trial!

Ln 510. This reader does not accept tumour volume can be measured with a precision greater than two decimal places (viz. 49.981%, 51.028%, and 8.0484%). This is simply unachievable in vivo!

Ln 528. Poor grammar. Please revise “The study measured serum levels of ALT and AST for n-hexane and ethyl acetate extracts” (sic)? For example, this should read ‘The study measured serum levels for ALT and AST… in animals exposed to/ injected/ treated… with C. oxyacantha extracts in n-hexane and ethyl acetate.’

Ln 542. This reader is concerned! How does an extract (i.e solute) “show significance”? This is non-sensical.

Ln 547. How can the precision of the error (3 decimal places) be greater than the accuracy of the assay result (2 decimal places)? This clearly illustrates that the authors do not understand the mathematics of basic statistics and the difference between precision and accuracy! Precision and accuracy are not the same thing!

Ln 569. For consistency, please replace “MDAMB231” with ‘MDA-MB231’?

Ln 573. What is the evidence “Most anticancer medications used nowadays in chemotherapy are cytotoxic to normal cells, leading to unwanted side effects”?

Ln 578. For consistency, please replace “MDAMB231” with ‘MDA-MB231’?

Ln 575. The statement “The current study provides a significant bearing in the search for compounds that can reduce the harmful side effects of anticancer drugs…” is overinterpretation. No part of this study examined side effects of associated with anti-neoplastic chemotherapies. Please revise?

Ln 580. As previously commented, the authors overinterpret caspase 3 assay data with “induction of apoptosis”. No evidence is provided to validate “induction of apoptosis” (e.g. a simple apoptosis assay). Caspase 3 is known to have other biological sequelae.

Ln 585. No data is reported to substantiate the claim “solvent extracts exhibited moderate to mild apoptotic effects” (sic).

Ln 589. What is “increased cure percentage” (sic)? 

What is the evidence for “observed apoptosis induction”?

Ln 601. What is the evidence “The results indicated that Carthamus oxycantha extracts exhibit a varying degree of increased proliferation in both T and B lymphocytes” (sic)? The reported data did not assay “both T and B lymphocytes”! To what data are the authors’ referring?

Ln 604. This reader is confused by relevance of the statement “Generally, several phytochemicals are studied for their immunomodulating and anticancer properties” (sic)? It is not clear to this reader how the following paragraphs (Ln 604 – Ln 634) have relevance to this report. No attempt is made to cross-reference, or compare with compounds characterised by the authors through LC-MS.

Ln 636. As it is clear the authors are not expert in clinical biochemistry, please provide substantiating evidence (citation) for “The enzymes responsible for liver and kidney function, particularly ALT, AST, and creatinine, are highly indicative of toxicity and safety levels at therapeutic doses due to their crucial role in drug metabolism and excretion.”

Ln 678. This reader is confused. Why is the “reduction in tumor size, can be considered a reliable depiction of the preventive impact of Carthamus oxyacantha” (sic)? I suggest that this is overreach.

Ln 679. The subsequent claim “The study also revealed that the plant's anticancer activity is attributed to multiple mechanisms of action synergistically enhanced by phytochemicals” (sic) is overinterpretation.

Ln 682. I am afraid that I cannot agree “…it has been established that the plant induces apoptosis and modulates the immune system.” (sic) The experimental data presented in this report does not substantiate this claim. The mechanisms of action have yet to be verified, nor have they been validated. They are most certainly not established! 

Ln 716. Ref 5. Incomplete data.

Ln 718. Ref 6. Incomplete data.

Ln 763. Ref 28. Incomplete data.

Ln 814. Ref 50. Incomplete data.

Comments on the Quality of English Language

While it is evident English is not the author’s native language, the text is generally adequate, although several instances will require sub-editorial intervention for grammar.

Author Response

Please see attached author response
